# Systemic vasoconstriction and mortality in patients with heart failure and reduced ejection fraction: A cohort of patients who underwent non-invasive hemodynamic monitoring

**Marcelo Eidi Ochiai** [ID]*[◔], **Kelly Regina Vieira Novaes**[◔], **Lucas Hideki Kato Myakava**[‡], **Marcelo Villaça Lima**[‡], **Euler Cristovan Ochiai Brancalhão**[‡], **Juliano Novaes Cardoso**[‡], **Solange de Sousa Andrade**[‡], **Wilson Jacob Filho**[‡], **Antonio Carlos Pereira Barretto**[◔]

Heart Institute (InCor), University of São Paulo, São Paulo City, SP, Brazil

◔ These authors contributed equally to this work.
‡ LHKM, MVL, ECOB, JNC, SSA and WJF also contributed equally to this work.
* marcelo.ochiai@incor.usp.br

## Abstract

Advanced heart failure primarily manifests during and after hospitalization for decompensation. Identifying prognostic factors is crucial for distinguishing patients who may benefit from drug therapy from those with end-stage disease. This study aimed to evaluate the prognostic significance of systemic vasoconstriction in patients with decompensated heart failure with a reduced ejection fraction. We evaluated patients hospitalized for decompensated heart failure with a left ventricular ejection fraction of < 40% who underwent non-invasive hemodynamic monitoring using the Modelflow method. The primary endpoint was all-cause mortality, and the data were analyzed using logistic regression. This study included 58 patients (71% men) with a mean age of 58.9 years, an ejection fraction of 23.4%, a median B-type natriuretic peptide of 1,005.0 pg/mL (interquartile range = 1,498.0), and 43% with Chagas disease. The cardiac index was 2.7 L·min$^{-1}$·m$^{-2}$, and the systemic vascular resistance index was 2,403.9 dyn·s·cm$^{-5}$·m$^{-2}$. Over an average follow-up of 29.0 months, 51 (87.9%) patients died. Assessing three-year mortality, high systemic vascular resistance indices were predictive of events with a relative risk of 3.9 (95% confidence interval = 1.1–13.9; *P*-value = 0.037). In conclusion, non-invasive hemodynamic monitoring identifies systemic vasoconstriction, which is associated with poor prognosis in patients with advanced heart failure and reduced ejection fraction.

## Introduction

Heart failure is an ancient problem described in the Gospel of Luke, the Physician, in Chapter 14, verses 2–4 [1]. Nowadays heart failure is a public health problem because of its increasing

**Data Availability Statement:** All relevant data are within the paper and its Supporting Information files.

**Funding:** This study was financially supported by São Paulo Research Foundation (FAPESP) [https://fapesp.br] in the form of a grant (2008/03460-4) received by ACPB. No additional external funding was received for this study.

**Competing interests:** The authors have declared that no competing interests exist.

prevalence. Decompensation management in patients with heart failure with reduced ejection fraction (HFrEF) requires a specific treatment that can differ significantly from that in patients with stable heart failure. Progress in HFrEF treatment, including new drugs, allows for an improvement in patient prognosis and a subsequent reduction in mortality; however, identifying patients who could benefit from these interventions is crucial. Therefore, prognostic factors are important in therapeutic planning for patients with HFrEF.

A low ejection fraction, hyponatremia, and renal dysfunction are classic prognostic factors for HFrEF, even during decompensation [2]. These factors change throughout the course of the disease, including during hospitalization and in response to optimized drug therapy. Additionally, other factors such as natriuretic peptides in the serum and urine can predict death or rehospitalization in patients with HFrEF [3].

Disease progression and increased neurohormonal activation result in a low cardiac output and systemic vasoconstriction. The pathophysiological importance of this process is demonstrated by the reduction in mortality associated with the use of vasodilators, such as angiotensin-converting enzyme (ACE) inhibitors [4], angiotensin receptor antagonists [5], and hydralazine/nitrates [6].

Invasive hemodynamic monitoring, such as pulmonary and peripheral artery catheters, has been used to guide cardiovascular therapy for the management of patients with decompensated heart failure [7]. Usually, the hemodynamic targets of HFrEF treatment are pulmonary capillary pressure < 15 mmHg and cardiac index > 2.0 L/min·m$^2$ [8]. Venous puncture accidents, thrombosis, and bloodstream infections related to the catheters limit the use of invasive monitoring.

Therefore, non-invasive hemodynamic monitoring is crucial for the care of patients with HFrEF. The ModelFlow method has been used as a non-invasive hemodynamic monitoring method. The ModelFlow method is based on the pulsatile output of the finger arterial walls using an inflatable finger cuff with a built-in photoelectric plethysmograph. The monitor calculates cardiac output while continuously measuring blood pressure [9] and this non-invasive monitoring can be used in cardiac anesthesia [10] and non-cardiac surgery [11]. The Modelflow method is also known as finger cuff monitoring.

This study aimed to assess the prognostic value of non-invasively identified systemic vasoconstriction in patients with decompensated HFrEF.

## Materials and methods

This study focused on a contemporary observational cohort of patients with HFrEF who were hospitalized in the acute HF unit of a tertiary public hospital associated with a public university. The inclusion criteria were as follows: hospitalization for decompensated HF with a left ventricular ejection fraction of < 40%, dyspnea or fatigue at rest, peripheral edema, pulmonary rales, or jugular vein distension. Patients with active, uncontrolled systemic infections as well as acute coronary syndromes were excluded from the study.

### Procedures

Upon admission, all patients were symptomatic at rest. Patients who showed symptom improvement after the initial use of diuretics, vasodilators, or intravenous inotropes were considered eligible for non-invasive hemodynamic monitoring. Patients were hemodynamically stable with or without intravenous inotropic drug infusion and were asymptomatic at rest, generally after 24 h of admission. Non-invasive hemodynamic monitoring was performed with patients lying supine on the hospital bed using specific equipment following the Modelflow method (Nexfin™, BMEYE, The Netherlands; now, ClearSight™, Edwards Lifesciences,

Irvine, CA). According to Ameloot et al. [12], "The Nexfin method is based on measuring arterial pressure using an inflatable cuff around the middle phalange of the finger. The pulsating finger artery was clamped to a constant volume by applying a varying counterpressure equivalent to the arterial pressure using a built-in photoelectric plethysmograph and an automatic algorithm (Physiocal). The resulting finger arterial pressure waveform was reconstructed into a brachial artery pressure waveform using a generalized algorithm. The cardiac index (NexCO) was calculated using the pulse contour method (CO-TREK), measured systolic pressure time integral, and heart afterload determined from the Windkessel model". The formulas are [13]:

$$V_Z = A_{SYS} \div Z_{AO}$$

$$V_{CZ} = V_Z[0.66 + 0.005 \times HR - 0.01 \times age \times (0.014 \times Pmean - 0.8)]$$

$$CO_{CZ} = V_{CZ} \times HR \times cal$$

$$SVRI = Pmean \div (CO_{CZ} \div BSA)$$

Where $V_Z$ is the stroke volume, $A_{sys}$ is the area under the systolic portion of the arterial pressure wave, $Z_{ao}$ is the aortic impedance, $V_{CZ}$ is the corrected stroke volume, HR is the heart rate, Pmean is the mean arterial pressure, $CO_{CZ}$ is the Wesseling's pulse contour cardiac output, the calibration factor is cal = $CO_{CZ}/CO_{ref}$, SVRI is the systemic vascular resistance index, and BSA is the body surface area.

Monitoring was performed for at least 5 min and was recorded in the internal memory of the equipment. The defined cardiac index and systemic vascular resistance index (SVRI) values were obtained after meticulous review of the more stable segments of the curve, referred to as the "revised curve." Patients were assigned to 1 of 2 groups, based on an SVRI cutoff of 1,200 dyn·s·cm$^{-5}$·m$^{-2}$: a higher and lower SVRI group [14].

After discharge from the hospital, the patients were followed up using our ambulatory service. A physician who was unaware of the study protocol prescribed the necessary medications. During hospitalization, drug therapy primarily involving beta-blockers was optimized. Typically, patients underwent re-evaluation 30 days after discharge and were subsequently followed-up for 6 months. Our institution aims to achieve optimized drug therapy in alignment with current guidelines during each medical appointment. The primary endpoint was all-cause mortality, and a log-rank test showed 75% mortality in the higher SVRI group compared to 50% in the lower SVRI group, providing a sample power of 81%.

## Statistical analysis

Variables are expressed as number and proportion, mean and standard deviation (SD) or median and interquartile range (IQR). Chi-squared or non-paired Student's t-test was used to compare the two groups, with a two-sided *P*-value < 0.05, considered statistically significant. Predictors of three-year all-cause mortality were defined using multivariate logistic regression [15], and expressed as relative risks and 95% confidence intervals (CIs). The following variables were analyzed using logistic regression: age, ejection fraction, hyponatremia, renal dysfunction, Chagas disease, cardiac index, and SVRI. We included hemodynamic variables in addition to those known to be related to mortality in heart failure. Survival curves were created using the Kaplan-Meier method and compared using the log-rank test [16].

### Ethics

Each participant provided a written informed consent upon admission. The study protocol conformed to the ethical guidelines of the 1975 Declaration of Helsinki and was approved by the local Research Ethics Committee. The authors declare no conflicts of interest regarding the Nexfin manufacturer.

## Results and discussion

Between 01 April 2009 and February 28, 2013, 196 patients were admitted to our hospital, of whom 127 underwent non-invasive hemodynamic monitoring. After revising the curves, 58 patients who attended the follow-up appointments were included in this study (Fig 1).

The non-invasive monitoring was done 19 days (median; IQR = 28 days) after admission. The median duration of non-invasive hemodynamic monitoring was 100 s (interquartile range [IQR] = 126), with 152 beats (IQR = 154). The selected segment for curve revision had a median of eight beats (IQR = 9) in 7 s (IQR = 6).

All included patients were admitted for resting dyspnea (New York Heart Association class IV). Of the 58 patients, 41 (71%) were men, with a mean age of 58.9 years, ejection fraction of 23.4% (SD = 5.2), and median B-type natriuretic peptide (BNP) of 1,005.0 pg/mL (IQR = 1,498); 25 (43%) of them had Chagas disease. During hospitalization, the following drugs were administered: beta-blockers, 43 (74%); ACE inhibitors, 30 (52%); angiotensin receptor blockers, 12 (21%); diuretics, 43 (74%); spironolactone, 12 (21%); and hydralazine, 30 (52%).

Dobutamine was administered to 46 (79%) patients during non-invasive hemodynamic monitoring. The cardiac index was 2.7 (SD = 0.7) L·min$^{-1}$·m$^{-2}$, and the median SVRI was 2,116.0 (IQR = 912) dyn·s·cm$^{-5}$·m$^{-2}$. Thirty-five (60%) patients with an increased SVRI

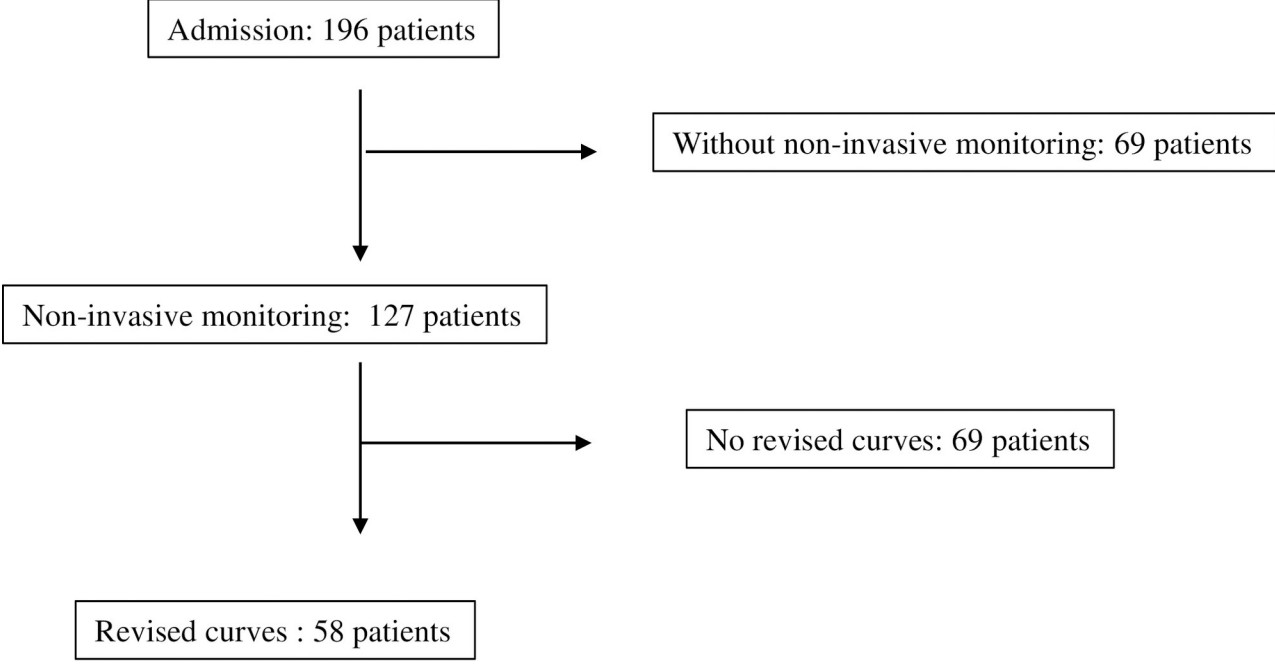

**Fig 1. Flow diagram.** Revised curves = the hemodynamic monitoring that could be selected a part with less oscillation and less interference to calculate cardiac index and systemic vascular resistance index.

**Table 1. Baseline characteristics according to hemodynamic pattern.**

| Variables | Higher SRVI (n = 35) | Lower SVRI (n = 23) | P-value |
|---|---|---|---|
| Age (years) | 57.0 (15.1) | 59.7 (11.6) | 0.483 |
| Ejection fraction (%) | 22.2 (4.6) | 25.6 (5.8) | 0.019 |
| Sodium (mEq/L) | 136 (5.7) | 135 (2.8) | 0.443 |
| Creatinine (mg/dL) | 1.56 (0.60) | 1.26 (0.25) | 0.013 |
| BNP—pg/mL | 1,416.5 (1,461.0) | 674.0 (1,345.0) | 0.023 |
| Cardiac index (L/min/m2) | 2.3 (0.6) | 3.3 (0.6) | < 0.001 |
| SVRI (dyn·s·cm$^{-5}$·m$^{-2}$) | 2,548.5 (1,050.0) | 1,772.0 (387.0) | < 0.001 |
| Beta-blocker (%) | 29 (83) | 16 (70) | 0.654 |
| ACE inhibitors (%) | 9 (26) | 12 (52) | 0.450 |
| ARB (%) | 3 (9) | 7 (30) | 0.450 |
| Hydralazine (%) | 22 (63) | 7 (30) | 0.097 |

All data formatted as median (IQR) or as mean (SD)

SVRI, systemic vascular resistance index; IQR, interquartile range; SD, standard deviation; ACE, angiotensin-converting enzyme; ARB, angiotensin receptor blocker.

presented with a more depressed ejection fraction, lower cardiac index, higher serum BNP levels, and worse renal function (Table 1).

Forty-four patients were discharged from hospital, of whom 30 (68.2%) were readmitted. Thirteen patients with high peripheral resistance were discharged, and among them, eight (61.5%) were readmitted. The readmission rate was similar to that in the low peripheral resistance group (*P* = 0.587).

Among the included patients, 51 died after a mean follow-up of 29 months, with 14 (24.1%) patients dying during their hospital stay (Tables 2 and 3). Pre-specified variables (age, ejection fraction, hyponatremia, BNP, renal dysfunction, Chagas disease, cardiac index, and SVRI) were subjected to a logistic regression analysis. When considering three-year mortality, higher SVRIs were a predictor of events with a relative risk of 3.88 (95% CI = 1.08–13.89; *P* = 0.037; Fig 2).

The remaining pre-specified variables, age (P = 0.07), ejection fraction (P = 0.748), Chagas disease (P = 0.376), BNP (P = 0.247), hyponatremia (P = 0.110), renal dysfunction (P = 0.711) and cardiac index (P = 0.926) were not predictors of mortality.

The primary finding of the present study was that non-invasively measured systemic vasoconstriction could be used to predict mortality in patients with HFrEF.

The incidence and prevalence of HF have increased, primarily in older populations. Symptom severity is particularly important in hospitalized patients with HFrEF, despite advances in drug therapy, which has had a dramatic impact on reducing mortality. Classic predictors of mortality in patients with HFrEF include low ejection fraction, renal dysfunction, and hyponatremia [2].

**Table 2. Mortality according to systemic vascular resistance index.**

| Variables | Higher SVRI (n = 35) | Lower SVRI (n = 23) | Odds ratio (95% CI) | P-value |
|---|---|---|---|---|
| One-year mortality | 22 (64.7%) | 10 (43.5%) | 2.38 (0.81–7.04) | 0.143 |
| Three-year mortality | 29 (85.3%) | 14 (60.9%) | 3.88 (1.08–13.89) | 0.037 |
| Total mortality | 31 (91.2%) | 20 (87.0%) | 1.55 (0.28–8.45) | 0.175 |

SVRI = systemic vascular resistance index; higher SVRI: > 1,200 dyn·s·cm$^{-5}$·m$^{-2}$; lower SVRI < 1,200 dyn·s·cm$^{-5}$·m$^{-2}$.

**Table 3. Baseline characteristics according to outcome.**

| Variables | Dead (n = 51) | Alive (n = 7) | *P-value* |
|---|---|---|---|
| Age (years) | 59.6 (13.1) | 45.2 (11.7) | 0.016 |
| Ejection fraction (%) | 24.1 (5.2) | 19.1 (3.1) | 0.004 |
| Sodium (mEq/L) | 135.9 (4.2) | 132.7 (7.2) | 0.293 |
| Creatinine (mg/dL) | 1.46 (0.51) | 1.21(0.39) | 0.174 |
| BNP—pg/mL; median (IQR) | 1,104 (1,633) | 679 (661) | 0.102 |
| Cardiac index (L/min/m2) | 2.71 (0.76) | 2.75 (0.66) | 0.902 |
| SVRI (dyn·s·cm$^{-5}$·m$^{-2}$) | 2,195.0 (1,017) | 2,028.0 (849) | 0.232 |

All data formatted as median (IQR) or as mean (SD)

SVRI, systemic vascular resistance index; IQR, interquartile range; SD, standard deviation.

An increase in neurohormonal activity occurs during the decompensation of HF, primarily in the renin-angiotensin-aldosterone, adrenergic, and natriuretic peptide systems, resulting in a subsequent increase in SVRI [17]. High SVRI indicates not only more severe heart failure but also an increase in cardiac workload. Consequently, a low cardiac index and high SVRI are indicative of mortality in patients with decompensated HFrEF [18]. Interestingly, in our study, high SVRI indicated worse outcomes but increased BNP and hyponatremia were not associated with mortality. The improved outcomes of patients treated with vasodilators, neurohormonal agents, or direct agents demonstrate the importance of peripheral vasoconstriction in HFrEF.

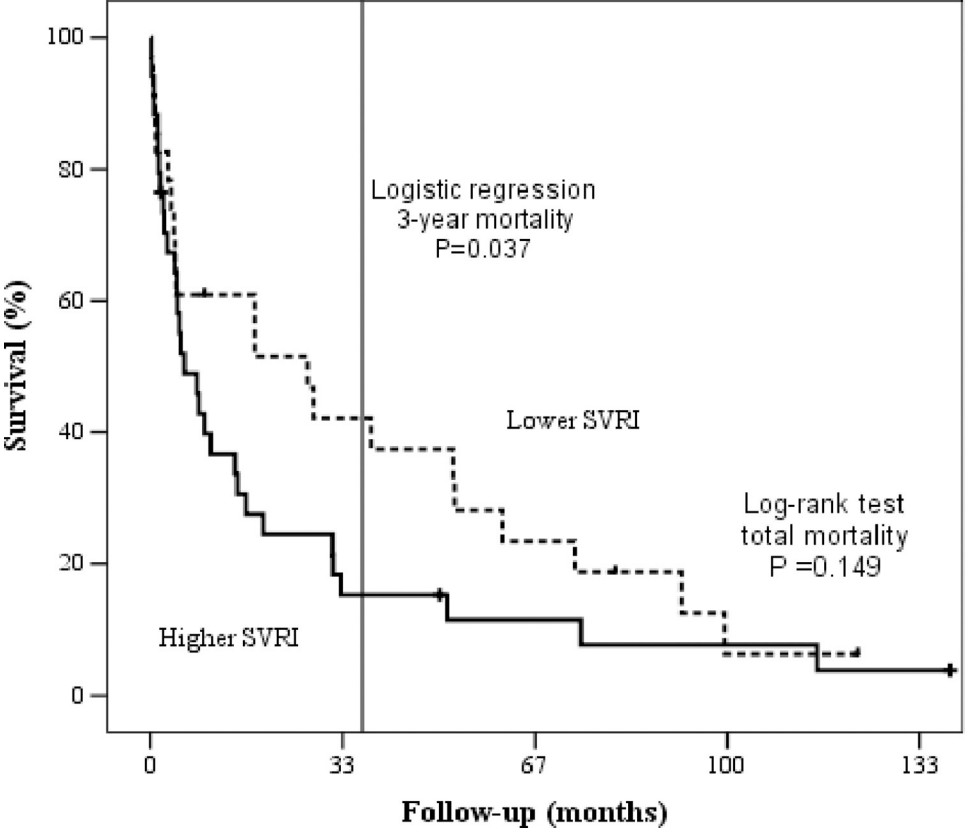

**Fig 2. Survival curves according to systemic vascular resistance index.** SVRI = systemic vascular resistance index.

The role of invasive hemodynamic monitoring in patients with advanced HFrEF has been previously investigated. Some studies have shown a divergence between clinical assessment and invasive hemodynamics [19] whereas others, such as the ESCAPE Trial, have demonstrated that invasive monitoring helps improve treatment strategies, despite the fact that it does not improve clinical outcomes [8]. Invasive hemodynamic monitoring is associated with complications, such as venous puncture accidents, catheter-related infections, and venous thrombosis.

The clinical hemodynamic profile provides an easy and helpful tool for managing decompensated HFrEF and identifying congestive and low cardiac output states [20]. This clinical assessment cannot distinguish between patients with more intense vasoconstriction and those with a more intense low cardiac output. Non-invasive hemodynamic monitoring could be useful in identifying severe vasoconstriction (high SVRI) that cannot be achieved through clinical assessment alone. The interplay among blood pressure, cardiac index, and systemic vascular resistance is complex. Consequently, high systemic vascular resistance may not result solely from low cardiac index. In our patient cohort, a low cardiac index did not correlate with mortality; however, high systemic vascular resistance did.

Various non-invasive methods have been developed to evaluate hemodynamics in patients with HFrEF, including the Modelflow method, impedance cardiography [21], and the partial carbon dioxide rebreathing technique [22]. The reliability of these non-invasive methods, compared with invasive measurements, has been previously studied. For instance, a study involving older patients hospitalized with HFrEF showed that invasive hemodynamic monitoring (using the PICCo method) is safe and associated with a shorter intensive care period than impedance cardiography [23]. Further explanations of the advantages and limitations of each method will enhance the clarity of their applicability to our study.

Briefly, the Modelflow method estimates the cardiac output based on a peripheral contour wave pulse. The Modelflow method showed a correlation coefficient of 0.55 when measuring the cardiac index [24]. De Wilde et al. [13] showed an agreement of 81% between the cardiac indices obtained from the Modelflow and invasive thermodilution methods. The ModelFlow method generates other hemodynamic variables with considerable variation [25]. The finger cuff sensor generates a hemodynamic curve with considerable oscillation from the baseline, which could interfere with the determination of the cardiac index and systemic vascular resistance. After recording the hemodynamic data, we selected a more stable monitoring segment to generate reliable data; however, this revision could not be performed in real time at the bedside. Regarding blood pressure, finger cuff sensors seem more sensitive in detecting systemic hypotension during digestive endoscopy than traditional oscillometric methods [26].

In a recent publication, a cohort of 257 patients with acute heart failure subjected to non-invasive monitoring illustrated that a hemodynamic profile featuring high systemic vascular resistance predicted a higher 90-day mortality than that predicted by the control group (10% vs. 3%) [27]. These findings are consistent with our findings. However, our study, with an extended follow-up duration, observed a significant number of fatal events, enhancing the statistical power.

Interestingly, the results of the present study showed a relationship between a higher SVRI and other hemodynamic variables, as well as decreased renal function and increased serum BNP levels. All these variables predicted a worse prognosis, indicating complex clinical, hemodynamic, and neurohormonal interactions in patients with acutely decompensated HFrEF.

Although the reliability of the non-invasive determination of SVRI is lower than expected, the identification of high-risk patients is crucial for clinical management [24]. Considering the limitations in the precision of this non-invasive method for evaluating vasoconstriction, strategies to minimize such inaccuracies should be explored. Addressing these limitations would

strengthen the reliability of the results and enhance the utility of this method for identifying patients who could benefit from additional vasodilatory drug therapy. Therefore, the measurement of SVRI through easy, non-invasive monitoring could be used to identify patients who could benefit from additional vasodilation drug therapy.

Our study findings on non-invasive hemodynamic monitoring and its association with outcomes in patients with HFrEF contribute valuable information to the existing literature. This novel insight may aid in refining therapeutic approaches and decision-making for the management of patients with HFrEF, thus providing potential benefits for patient outcomes and overall care [28].

## Study limitations

The present study has some limitations. The sample size of the study was small and included patients with advanced heart failure, characterized by an uncommonly high mortality rate, likely attributable to the severity of the heart disease. These individuals were from an older demographic who were prescribed the best recommended drug therapy at that time. Non-invasive hemodynamic measurements differ from the thermodilution method and allow for greater variability in each cardiac beat. The protocol used in the present study did not interfere with the drug therapy prescribed by the cardiologist. Additionally, the findings of the present study cannot be extrapolated to patients with HF and preserved ejection fraction.

## Conclusion

Non-invasive hemodynamic monitoring identifies systemic vasoconstriction, which is associated with poor prognosis in patients with advanced heart failure and reduced ejection fraction.

## Supporting information

**S1 Raw data. The datasets generated for the study.**
(XLSX)

## Acknowledgments

We thank the multidisciplinary team that took care of the patients and helped input the study data. We would like to thank Editage (www.editage.com) for the English language editing.

## Author Contributions

**Conceptualization:** Marcelo Eidi Ochiai, Marcelo Villaça Lima, Antonio Carlos Pereira Barretto.

**Data curation:** Kelly Regina Vieira Novaes, Lucas Hideki Kato Myakava, Marcelo Villaça Lima, Euler Cristovan Ochiai Brancalhão.

**Formal analysis:** Marcelo Eidi Ochiai, Kelly Regina Vieira Novaes, Lucas Hideki Kato Myakava, Marcelo Villaça Lima, Euler Cristovan Ochiai Brancalhão, Juliano Novaes Cardoso.

**Funding acquisition:** Marcelo Eidi Ochiai, Marcelo Villaça Lima, Antonio Carlos Pereira Barretto.

**Investigation:** Marcelo Eidi Ochiai, Marcelo Villaça Lima, Euler Cristovan Ochiai Brancalhão, Juliano Novaes Cardoso.

**Methodology:** Marcelo Eidi Ochiai, Kelly Regina Vieira Novaes, Lucas Hideki Kato Myakava, Marcelo Villaça Lima.

**Project administration:** Antonio Carlos Pereira Barretto.

**Resources:** Marcelo Eidi Ochiai, Antonio Carlos Pereira Barretto.

**Supervision:** Solange de Sousa Andrade, Wilson Jacob Filho, Antonio Carlos Pereira Barretto.

**Validation:** Kelly Regina Vieira Novaes, Solange de Sousa Andrade.

**Writing – original draft:** Marcelo Eidi Ochiai, Solange de Sousa Andrade, Wilson Jacob Filho, Antonio Carlos Pereira Barretto.

**Writing – review & editing:** Marcelo Eidi Ochiai.

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
