## [Decision Letter · Decision Letter 0]

9 Jun 2024

PONE-D-24-10329Systemic vasoconstriction and mortality in patients with heart failure and reduced ejection fraction: A cohort of patients who underwent non-invasive hemodynamic monitoringPLOS ONE

Dear Dr. OCHIAI,

Thank you for submitting your manuscript to PLOS ONE. After careful consideration, we feel that it has merit but does not fully meet PLOS ONE’s publication criteria as it currently stands. Therefore, we invite you to submit a revised version of the manuscript that addresses the points raised during the review process.

**ACADEMIC EDITOR: **We have now received comments from two experts in the field who have carefully reviewed your work. While they acknowledge the significance of your study and appreciate the thoroughness of your research, both reviewers have raised several critical issues that need to be addressed before we can proceed with the publication process.

Based on their feedback, we are requesting that you undertake a major revision of your manuscript.

We look forward to receiving your revised manuscript.

Kind regards,

Yashendra Sethi

Academic Editor

PLOS ONE

“São Paulo Research Foundation (FAPESP),

grant # 2008/03460-4

https://fapesp.br/en”

Additional Editor Comments: The methodlogy has concerns which puts conclusions made in question. Please address all these comments in detail and make required changes.

Reviewers' comments:

Reviewer's Responses to Questions

**Comments to the Author**

1. Is the manuscript technically sound, and do the data support the conclusions?

Reviewer #1: Partly

Reviewer #2: Yes

2. Has the statistical analysis been performed appropriately and rigorously? 

Reviewer #1: Yes

Reviewer #2: N/A

3. Have the authors made all data underlying the findings in their manuscript fully available?

Reviewer #1: No

Reviewer #2: No

4. Is the manuscript presented in an intelligible fashion and written in standard English?

Reviewer #1: Yes

Reviewer #2: Yes

5. Review Comments to the Author

Reviewer #1: In this article, the authors analyzed the use of Medflow (non invasive hemodynamic monitoring) to evaluate the systemic vascular resistance and determine if it predicts mortality in patients with ADHF.

We have some doubts about methodology:

1- How the SVRi was calculated (there is no mention of de CVP in the formula used). In the article it is mentioned that the SVRi was calculated by dividing the mean arterial pressure by the cardiac index.

2- In the baseline there is no mention about the use of inotropic drugs and if the acquisition of RVRi was priorily the introduction of inotropic

3- We think it is more appropriated to show the results in the final table (table2) divided in the two groups priorly analyzed (RVP <1200 and >1200). In the article the final analyses it was divided by outcome

Reviewer #2: I had the pleasure of reviewing the manuscript titled "Systemic Vasoconstriction and Mortality in Patients with Heart Failure and Reduced Ejection Fraction: A Cohort of Patients Who Underwent Non-Invasive Hemodynamic Monitoring." The manuscript effectively discusses and evaluates the prognostic significance of non-invasively identified systemic vasoconstriction in patients with decompensated heart failure and reduced ejection fraction (HFrEF). It should be free of grammatical and typographical errors. For instance, on line 141, the word "according" is missing a "to" after it. I recommend that the manuscript be reviewed for similar mistakes.

6. PLOS authors have the option to publish the peer review history of their article (what does this mean?). If published, this will include your full peer review and any attached files.

Reviewer #1: No

Reviewer #2: **Yes: **Abdelrahman Gad

---

## [Author Response · Author response to Decision Letter 0]

11 Jul 2024

Questions of Reviewer #1: “In this article, the authors analyzed the use of Medflow (non invasive hemodynamic monitoring) to evaluate the systemic vascular resistance and determine if it predicts mortality in patients with ADHF.

We have some doubts about methodology:

 How the SVRi was calculated (there is no mention of de CVP in the formula used). In the article it is mentioned that the SVRi was calculated by dividing the mean arterial pressure by the cardiac index.”

Answer: The formulas are:

V_Z = A_SYS÷Z_AO 

V_CZ =V_Z [0.66 +0.005 ×HR -0.01 ×age × (0.014 ×Pmean -0.8)]

〖CO〗_CZ =V_CZ ×HR ×cal

SVRI =Pmean ÷ (〖CO〗_CZ÷BSA)

Where VZ is the stroke volume, Asys is the area under the systolic portion of the arterial pressure wave, Zao is the aortic impedance, VCZ is the corrected stroke volume, HR is the heart rate, Pmean is the mean arterial pressure, COCZ is the Wesseling's pulse contour cardiac output, the calibration factor is cal=COCZ/COref, SVRI is the systemic vascular resistance index, and BSA is the body surface area.

(Lines 88 to 96)

Question: “In the baseline there is no mention about the use of inotropic drugs and if the acquisition of RVRi was priorily the introduction of inotropic.”

Forty-six patients (79%) received dobutamine during hemodynamic monitoring, as highlighted in the line 132.

 “We think it is more appropriated to show the results in the final table (table2) divided in the two groups priorly analyzed (RVP <1200 and >1200). In the article the final analyses it was divided by outcome.”

Answer:

Table 2. Mortality according to systemic vascular resistance indexed

Variables Higher SVRI 

(n = 35) Lower SVRI 

(n = 23) Odds ratio (95% CI) P-value

One-year mortality 22 (64.7%) 10 (43.5%) 2.38 (0.81-7.04) 0.143

Three-year mortality 29 (85.3%) 14 (60.9%) 3.88 (1.08-13.89) 0.037

Total mortality 31 (91.2%) 20 (87.0%) 1.55 (0.28-8.45) 0.175

SVRI = systemic vascular resistance index; higher SVRI: >1,200 dyn∙s∙cm−5∙m−2; lower SVRI <1,200 dyn∙s∙cm−5∙m−2.

Question of Reviewer #2: “I had the pleasure of reviewing the manuscript titled "Systemic Vasoconstriction and Mortality in Patients with Heart Failure and Reduced Ejection Fraction: A Cohort of Patients Who Underwent Non-Invasive Hemodynamic Monitoring." The manuscript effectively discusses and evaluates the prognostic significance of non-invasively identified systemic vasoconstriction in patients with decompensated heart failure and reduced ejection fraction (HFrEF). It should be free of grammatical and typographical errors. For instance, on line 141, the word "according" is missing a "to" after it. I recommend that the manuscript be reviewed for similar mistakes”.

Answer: We thank the reviewer for carefully reading and thereafter correcting the grammatical errors made in our manuscript.

---

## [Decision Letter · Decision Letter 1]

22 Aug 2024

PONE-D-24-10329R1Systemic vasoconstriction and mortality in patients with heart failure and reduced ejection fraction: A cohort of patients who underwent non-invasive hemodynamic monitoringPLOS ONE

Dear Dr. OCHIAI,

Thank you for submitting your manuscript to PLOS ONE. After careful consideration, we feel that it has merit but does not fully meet PLOS ONE’s publication criteria as it currently stands. Therefore, we invite you to submit a revised version of the manuscript that addresses the points raised during the review process.

We look forward to receiving your revised manuscript.

Kind regards,

Yashendra Sethi

Academic Editor

PLOS ONE

Reviewers' comments:

Reviewer's Responses to Questions

**Comments to the Author**

1. If the authors have adequately addressed your comments raised in a previous round of review and you feel that this manuscript is now acceptable for publication, you may indicate that here to bypass the “Comments to the Author” section, enter your conflict of interest statement in the “Confidential to Editor” section, and submit your "Accept" recommendation.

Reviewer #1: All comments have been addressed

Reviewer #2: All comments have been addressed

Reviewer #3: (No Response)

2. Is the manuscript technically sound, and do the data support the conclusions?

Reviewer #1: Yes

Reviewer #2: Yes

Reviewer #3: Partly

3. Has the statistical analysis been performed appropriately and rigorously? 

Reviewer #1: Yes

Reviewer #2: N/A

Reviewer #3: No

4. Have the authors made all data underlying the findings in their manuscript fully available?

Reviewer #1: Yes

Reviewer #2: Yes

Reviewer #3: Yes

5. Is the manuscript presented in an intelligible fashion and written in standard English?

Reviewer #1: Yes

Reviewer #2: Yes

Reviewer #3: Yes

6. Review Comments to the Author

Reviewer #1: In this article, the authors analyzed de use of Medflow (non invasive hemodynamic monitoring) to evaluate the systemic vascular resistance and determine if it predicts mortality in patients with ADHF. The authors responded adequately to the questions requested.

Reviewer #2: I recommend that the manuscript titled "Systemic Vasoconstriction and Mortality in Patients with Heart Failure and Reduced Ejection Fraction: A Cohort of Patients Who Underwent Non-Invasive Hemodynamic Monitoring" be accepted for publication in PLoS One. The authors have addressed previous comments comprehensively and have corrected all grammatical and typographical errors.

Reviewer #3: The authors have assessed the utility of a non-invasive hemodynamic method to measure peripheral arterial resistance and to establish its relationship with mortality. This is an interesting study that might draw the attention of readers. However I have some comments to improve the quality of the manuscript.

1. Please add reference, page 3, lines 51 and 52. Mortality reduction with ACEi and hydralazine/nitrates.

2. Did the authors exclude patients with acute coronary syndromes as the cause of HF decompensation? Were there any cases of acute myocardial infarction?

3. The authors should clarify the exact moment when the hemodynamic evaluation was done. As I understood, it was done at admission, after the patients received the initial treatment and became stable. Is that correct? All patients underwent the hemodynamic evaluation within 24 h of admission?

4. Statistics: It is not clear whether the authors did multivariate analysis. Was this multivariate logistic regression? Why did the authors use logistic regression? Since they constructed survival curves, I assume they have the date in which the events occurred. I believe it would be more appropriate to use as an endpoint the time to the event of mortality and use Cox proportional hazard models (univariate and multivariate analysis) to assess whether vasoconstriction was independently associated with mortality. This is a small sample but the number of events is elevated, allowing for multivariate analysis.

5. I understood that this is a retrospective study since patients were included back in 2009. If so, please add this information in methods, line 68.

6. In table 3, we see that patients who died had higher LVEF% (24.1 vs 19.1%). This was unexpected. Did the authors find an explanation for that? Perhaps it was by chance?

7. Please add p value in figure 2.

8. Please add in the limitations that this is a small sample.

9. Please comment in the discussion that in the ESCAPE trial, adjusting medications by hemodynamic profiles in patients with advanced HF did not reduce events as compared with clinical assessment alone. In addition, hemodynamics have been outperformed by neuro-hormonal assessment. So it is important to show in the manuscript that vasoconstriction was a predictor of mortality regardless of BNP.

7. PLOS authors have the option to publish the peer review history of their article (what does this mean?). If published, this will include your full peer review and any attached files.

Reviewer #1: No

Reviewer #2: **Yes: **Abdelrahman Gad

Reviewer #3: No

---

## [Author Response · Author response to Decision Letter 1]

6 Sep 2024

Thank you for carefully reviewing our manuscript and providing suggestions. We have provided our responses below. We hope that our answers and corrections based on these suggestions meet your expectations.

1) We have added a reference at the relevant location.

2) Since the study was conducted in a heart failure unit, patients with angina, acute ischemic electrocardiographic changes and increased cardiac enzymes were not included as they are treated in the coronary care unit of our institution. To be clear, we included acute coronary syndromes as an exclusion criterion (lines 72-73). 

3) We evaluated decompensation of heart failure after initial stabilization when the patient had minimal effort or no intravenous inotrope dependence. The hemodynamic monitoring was done 19 days (median) after admission with IQR of 28 days. In fact, all patients underwent the hemodynamic monitoring after 24 h of admission (line 78).

4) We used the logistic regression as multivariate analysis. Initially, we used Cox regression; however, we did not find difference between the groups’ higher and lower systemic vascular resistance. We attribute that finding to our long follow-up period, as most patients died, and hence, collapsing curves and a type II error was generated. The survival curves have shown a visual difference between the groups with more intensity at middle of the follow-up period. Therefore, we decided to analyze mortality at 1-year, 2-years, and 3-years post initial admission with logistic regression, which demonstrated a statistical significance (P=0.037) between the groups. 

5) The sample population of this study was assembled at beginning of the research protocol and we followed the patients all these years; therefore, ours is a prospective study (Fletcher and Fletcher, “Clinical Epidemiology: the essentials” chapter 5).

6) We believe that probably this finding (patients who died had higher LVEF% ) was by chance because of the small number of living patients at the end of follow-up period, although this was not statistical significant after multivariate logistic regression. 

7) To provide more information, we have added P-value by log-rank test of total follow-up and P-value of 3-year mortality by logistic regression in Figure2.

8) Thank you for the valuable suggestion. We have mentioned small sample size as a limitation of the study (line 226).

9) We have included this aspect of invasive hemodynamic monitoring in the discussion. The changes are in lines 150, 152-155, 172- 173.

---

## [Decision Letter · Decision Letter 2]

10 Oct 2024

Systemic vasoconstriction and mortality in patients with heart failure and reduced ejection fraction: A cohort of patients who underwent non-invasive hemodynamic monitoring

PONE-D-24-10329R2

Dear Dr. OCHIAI,

We’re pleased to inform you that your manuscript has been judged scientifically suitable for publication and will be formally accepted for publication once it meets all outstanding technical requirements.

Kind regards,

Yashendra Sethi

Academic Editor

PLOS ONE

Additional Editor Comments (optional):

Reviewers' comments:

Reviewer's Responses to Questions

**Comments to the Author**

1. If the authors have adequately addressed your comments raised in a previous round of review and you feel that this manuscript is now acceptable for publication, you may indicate that here to bypass the “Comments to the Author” section, enter your conflict of interest statement in the “Confidential to Editor” section, and submit your "Accept" recommendation.

Reviewer #1: All comments have been addressed

Reviewer #3: All comments have been addressed

2. Is the manuscript technically sound, and do the data support the conclusions?

Reviewer #1: (No Response)

Reviewer #3: Yes

3. Has the statistical analysis been performed appropriately and rigorously? 

Reviewer #1: (No Response)

Reviewer #3: Yes

4. Have the authors made all data underlying the findings in their manuscript fully available?

Reviewer #1: (No Response)

Reviewer #3: Yes

5. Is the manuscript presented in an intelligible fashion and written in standard English?

Reviewer #1: (No Response)

Reviewer #3: Yes

6. Review Comments to the Author

Reviewer #1: (No Response)

Reviewer #3: The authors have addressed all issues. The manuscript is now clear and the limitations have been pointed out. No further comments.

7. PLOS authors have the option to publish the peer review history of their article (what does this mean?). If published, this will include your full peer review and any attached files.

Reviewer #1: No

Reviewer #3: No

---

## [Editor Report · Acceptance letter]

9 Dec 2024

PONE-D-24-10329R2 

PLOS ONE

Dear Dr. Ochiai, 

I'm pleased to inform you that your manuscript has been deemed suitable for publication in PLOS ONE. Congratulations! Your manuscript is now being handed over to our production team.

Kind regards, 

on behalf of

Dr. Yashendra Sethi 

Academic Editor

PLOS ONE